# Comparison of Force Distribution during Laryngoscopy with the C-MAC D-BLADE and Macintosh-Style Blades: A Randomised Controlled Clinical Trial

**DOI:** 10.3390/jcm13092623

**Published:** 2024-04-29

**Authors:** Axel Schmutz, Ingo Breddin, Ramona Draxler, Stefan Schumann, Johannes Spaeth

**Affiliations:** 1Department of Anaesthesiology and Critical Care, Medical Centre, Faculty of Medicine, University of Freiburg, 79106 Freiburg, Germanyjohannes.spaeth@uniklinik-freiburg.de (J.S.); 2KARL STORZ SE & Co. KG, 78532 Tuttlingen, Germany

**Keywords:** laryngoscopy, videolaryngoscopy, laryngoscopes, airway management, C-MAC, D-BLADE, force measurement

## Abstract

**Background**: The geometry of a laryngoscope’s blade determines the forces acting on the pharyngeal structures to a relevant degree. Knowledge about the force distribution along the blade may prospectively allow for the development of less traumatic blades. Therefore, we examined the forces along the blades experienced during laryngoscopy with the C-MAC D-BLADE and blades of the Macintosh style. We hypothesised that lower peak forces are applied to the patient’s pharyngeal tissue during videolaryngoscopy with a C-MAC D-BLADE compared to videolaryngoscopy with a C-MAC Macintosh-style blade and direct laryngoscopy with a Macintosh-style blade. Beyond that, we assumed that the distribution of forces along the blade differs depending on the respective blade’s geometry. **Methods**: After ethical approval, videolaryngoscopy with the D-BLADE or the Macintosh blade, or direct laryngoscopy with the Macintosh blade (all KARL STORZ, Tuttlingen, Germany), was performed on 164 randomly assigned patients. Forces were measured at six positions along each blade and compared with regard to mean force, peak force and spatial distribution. Furthermore, the duration of the laryngoscopy was measured. **Results**: Mean forces (all *p* < 0.011) and peak forces at each sensor position (all *p* < 0.019) were the lowest with the D-BLADE, whereas there were no differences between videolaryngoscopy and direct laryngoscopy with the Macintosh blades (all *p* > 0.128). With the D-BLADE, the forces were highest at the blade’s tip. In contrast, the forces were more evenly distributed along the Macintosh blades. Videolaryngoscopy took the longest with the D-BLADE (*p* = 0.007). **Conclusions**: Laryngoscopy with the D-BLADE resulted in significantly lower forces acting on pharyngeal and laryngeal tissue compared to Macintosh-style blades. Interestingly, with the Macintosh blades, we found no advantage for videolaryngoscopy in terms of force application.

## 1. Introduction

Videolaryngoscopy has become one of the most essential techniques for airway management and is expected to gain even more clinical relevance in the future [1]. This forecast is supported by the accumulating evidence, attributing an improved view of the larynx [2], lower incidence of a postoperative sore throat [3], improved first-pass success [4,5,6], decreased number of attempts needed [7] and a great teaching potential to videolaryngoscopy [8]. It is less evident, however, how exactly the different videolaryngoscopes perform [1]. By rendering a direct view of the larynx unnecessary, video technology offers the chance to rethink the geometry of blades for laryngoscopy. Given this potential, blades of strong curvature have been developed, generating less tissue displacement and therefore reducing the risk of airway trauma [9]. Understanding these advantages requires differentiated knowledge of the applied forces to the pharyngeal and laryngeal tissue. In this regard, the evidence mainly results from studies with manikins [10,11,12,13]. Studies performed in humans, however, reveal the limitations of a simulated setup in this respect, as they refer to considerably lower forces [14,15]. Moreover, all previous human studies report accumulative force [16]. Such general information may help to estimate the risk of trauma during laryngoscopy but insufficiently supports the re-engineering of a blade’s geometry. Furthermore, the C-MAC D-BLADE (KARL STORZ SE & Co. KG, Tuttlingen, Germany) is one of the most commonly used hyperangulated blades for videolaryngoscopy, but only scarce evidence is available regarding force transmission during laryngoscopy with this device.

In order to understand the interdependency of a laryngoscope’s curvature and its impact on human airways, spatial resolution force measurements are required. Such information will provide insights into the force transmission during laryngoscopy and may also be of interest for the development of less traumatic blades.

We hypothesised that lower peak forces are applied to the patient’s pharyngeal and laryngeal tissue during videolaryngoscopy with a C-MAC D-BLADE compared to videolaryngoscopy with a C-MAC Macintosh-style blade and direct laryngoscopy with a Macintosh-style blade. Beyond that, we assumed that the distribution of forces along the blade differs depending on the respective blade’s geometry. Therefore, we measured the forces applied to the patient’s upper airways at six different positions along the respective blades during laryngoscopy in human patients.

## 2. Materials and Methods

This randomised controlled trial adheres to the Consolidated Standards of Reporting Trials (CONSORT) statement [17].

### 2.1. Participants and Randomisation

After ethical approval for this study (Ethical Committee, Freiburg University, #161/17, date of approval 18 May 2017; German Clinical Trials Register #DRKS00012841) and having gained written informed consent, 164 adult patients requiring endotracheal intubation for elective otorhinolaryngology surgery at the University Medical Centre Freiburg were randomly assigned to either videolaryngoscopy with a D-BLADE or C-MAC or direct laryngoscopy with a Macintosh blade (all KARL STORZ SE & Co. KG, Tuttlingen, Germany) following a defined sequence. Due to the technical requirement of hygienic processing and subsequent sensor calibration, randomisation was based on a block-randomisation procedure: on each day of measurement, two sets of each laryngoscope type were prepared. The randomisation sequence (drawn via www.random.org on a daily base, author AS) included the order of the six prepared laryngoscopes to be used in the consecutive patients, respectively, on this day. Only patients with physical status 1 or 2 according to the American Society of Anesthesiologists were included. Patients with expected difficult intubation (Mallampati class > 3, interincisor distance < 3 cm, thyromental distance < 6 cm) [18], gastro-oesophageal reflux disease or relevant cardiac or pulmonary comorbidities were excluded from the study. Patients’ characteristics and airway evaluation including the Arné risk index score [19], a multivariable risk index score for the preoperative assessment of difficult intubation with direct laryngoscopy, were recorded. An Arné risk index score of 11 or greater identifies difficult tracheal intubation, with a sensitivity of 94 per cent and specificity of 96 per cent in general surgery patients.

### 2.2. Laryngoscopy Blades and Methods of Measurements

All the blades were intended for laryngoscopy of adult patients, with the Macintosh-style blades being of comparable size (size 4) but with clear differences in curvature (Figure 1). In order to measure the forces arising along the blades’ profiles, circular tactile force sensors (FlexiForce HT201, Tekscan Inc., Boston, MA, USA) were used. This sensor type provides reliable spatial and temporal resolution and has already been established in a comparable context [14]. According to information from the manufacturer, the sensor provides a measuring range of up to 133 N with an error of ±3.3% at sufficient linearity. A single sensor string has a length of 197 mm, a width of 14 mm and a thickness of only 0.2 mm. The circular-shaped sensing area at the string’s end has a cross-sectional diameter of 9.5 mm, resulting in a force sensing area of 71 mm^2^. Our own preliminary tests on an airway manikin showed that six adjacent sensors sufficiently captured the part of the blade in direct contact with the airway. Accordingly, for the clinical study, six sensors were glued to the concave surface of each blade, equidistantly and in an aligned fashion, using LOCTITE M-31-CLTM medical device adhesive (Henkel Corporation, Düsseldorf, Germany; Figure 2). 

In order to provide optimal transmission of the pressure, circular-shaped biocompatible silicone elastomers (SILPURAN, Wacker, Munich, Germany), with a thickness of 1.5 mm and a diameter corresponding to that of the sensing area, were attached to the top of the sensors’ surfaces, as recommended by previous research [14]. In order to shield the sensors’ surfaces from contamination by secretions and mechanical damage, the blades were covered with a transparent adhesive film dressing (Hydrofilm, Paul Hartmann, Berlin, Germany). After each laryngoscopy, the adhesive film was removed and the respective blade (including the sensors) underwent an automated cleaning and disinfection process using glutaraldehyde, following departmental routine. The sensors were connected to an analogue–digital converter (CEBO-MSA64, CESYS, Herzogenaurach, Germany) using a flat flexible cable which was guided by an assistant during laryngoscopy. The analogue–digital converter automatically calculates the force from the measured pressure and the given cross-sectional area of the sensors’ surface. The data were recorded simultaneously for the aligned sensors at a sample rate of 50 Hz. The sensors were calibrated before each measurement. The respective blade was clamped into a specific calibration system (Figure 2) to perform a multipoint calibration before each measurement. After calibration to the atmosphere, defined forces were orthogonally applied to each respective sensor in steps of 5 N up to 30 N, using a force gauge (DFX-II series, AMETEK Chatillon Scales, Largo, FL, USA). Pressure measuring points are referred to as S1 to S6 in consecutive order from the tip to the base of the blade.

### 2.3. Anaesthesia and Study Protocol

After implementing routine hemodynamic and respiratory monitoring, general anaesthesia was induced, following our clinical standard approach. This included continuous intravenous infusion of remifentanil at 0.3 µg kg^−1^ min^−1^ (ideal body weight), started 5 min in advance to target-controlled infusion of propofol with an effect-site concentration of 5.0 µg mL^−1^ (Schnider-Model). A neuromuscular blockade was induced using cisatracurium besilate at 0.08 mg kg^−1^ ideal body weight, and conditions for laryngoscopy were considered sufficient 3 min thereafter. The patients’ heads were placed in the “sniffing position” on a pillow 5 cm high. 

All the laryngoscopies were conducted by anaesthetists with experience of more than 100 tracheal intubations. The anaesthetists were asked to intubate the trachea for the best achievable view of the larynx and to withdraw the blade afterwards. Force measurements were started just before the introduction of the blade in the patient’s oral cavity and stopped just after the blade had been completely withdrawn. The participating anaesthetists were blinded to the purpose of the study and the pressure measurement values. The data were analysed offline with computational algorithms using MATLAB (The MathWorks, Inc., Natick, MA, USA), and the data of all the measurements were included. Hence, all the data were treated equally. Data processing was programmed by an engineer not involved in the clinical data assessment (author SS). Each measurement was visually checked for completeness. Four variables were calculated based on the clinical pressure measurements: (1) the mean force across all sensors during the intubation interval was calculated for comparison of average effective forces; (2) the peak force of each single sensor during the intubation interval was calculated for comparison of local maximum forces; (3) the mean impulse force as the averaged force across all sensors in the first third of the laryngoscopy interval ($I=\int_{t=0}^{t=33\%}F\;(t)\;\cdot dt$) was calculated to determine the temporal dependency of average effective forces; and, (4) the time interval required for laryngoscopy (determined as the time between the first and the last recorded sensor signal) was calculated for comparison of the duration of laryngoscopy required.

### 2.4. Sample Size Calculation and Statistical Analysis

For the sample size calculation, we conducted a preliminary (unpublished) manikin study in a crossover design including 30 experienced anaesthetists performing laryngoscopy with each of the included devices. In this study, we determined a mean force of 5.6 N and a mean standard deviation of 3.6 N (on each sensor). We estimated an effect size of 2.1 N (referring to a cumulative force difference of 12.6 N on all six sensors in total) as relevant. Allowing for a type one error of 0.05 in the post hoc *t*-test, a sample size of 48 subjects per group was required to achieve a test power of 0.80. To compensate for potential dropouts, the sample size was increased to 52 subjects per group. The data were statistically processed using GraphPad Prism 9.1.0 (GraphPad Software Inc, La Jolla, CA, USA). The normal distribution of continuous variables was tested using the Kolmogorov–Smirnov test. The values for mean force and peak force showed normal distribution and were subsequently compared using an ordinary one-way analysis of variance (ANOVA), followed by Tukey’s multiple comparisons as a post hoc test. The values for impulse force and duration of laryngoscopy were not normally distributed and were therefore compared using a Kruskal–Wallis test. The values for the peak force of isolated sensors along a particular blade were compared by one-way repeated measures ANOVA. In order to compare the values for peak forces of isolated sensors of respective positions between different blades, ordinary one-way ANOVA and post hoc Tukey’s multiple comparisons were calculated. A *p*-value of <0.05 was considered significant. Results for mean force, peak force, impulse force and time for laryngoscopy are given as the median andinterquartile range IQR (25–75).

## 3. Results

In total, 432 patients were assessed for eligibility. Of those, 257 patients did not match the inclusion criteria, and 11 patients refused to participate. Thus, 164 patients were randomised for intervention (Figure 3). 

The data from all 164 patients included was analysed. Table 1 summarises the patients’ characteristics and scores for airway evaluation.

With the D-BLADE, the mean force was, on average, 0.9 N (52%), peak force was 2.2 N (35%), and impulse force was 15.4 N (57%) lower compared to the Macintosh blades during video- or direct laryngoscopy (Figure 4; all *p* < 0.011). There were no differences between the Macintosh blades for any of these variables (all *p* > 0.615). 

Regarding their spatial distribution, the peak forces differed considerably between adjacent force-measuring sensors in all the blades (Figure 5 upper panel; ANOVA p_Position_ < 0.0001). Moreover, the peak forces differed between blades at sensors of comparable positions (Figure 5; p_Blade_ < 0.005). Peak forces were lower with the D-BLADE at each respective sensor position compared to the Macintosh blades during videolaryngoscopy (all *p* < 0.009) and direct laryngoscopy (all *p* < 0.019). There were no differences in the peak forces at any position between the Macintosh blades (all *p* > 0.128). With the D-BLADE, the maximum forces were measured at the sensor closest to the blade’s tip (S1, 2.71 (1.28–4.77) N). By contrast, with the Macintosh blades, the highest forces were recorded at the second sensor (S2) during videolaryngoscopy (4.83 (2.99–6.24) N) and direct laryngoscopy (5.14 (3.14–6.18) N). For all the blades, the lowest peak forces were recorded at the sensor closest to the handle.

With the D-BLADE (34.2 ± 23.1 s), laryngoscopy took longer compared to videolaryngoscopy (24.7 ± 15.6 s, *p* = 0.02) and direct laryngoscopy (25.2 ± 11.5 s, *p* = 0.014) with the Macintosh blades. The duration of laryngoscopy did not differ between the Macintosh blades (*p* > 0.99). Figure 6 shows the peak forces on the isolated force measurement sensors over time.

## 4. Discussion

During laryngoscopy with a D-BLADE, less force was exerted on the pharyngeal and laryngeal tissue compared to laryngoscopy with a Macintosh-style blade. The average force exerted during laryngoscopy with the D-BLADE was only about half of the force exerted with the straighter blades. These findings are in accordance with earlier studies evaluating hyperangulated blades from other manufacturers [14]. In a simulated setup, experienced providers exerted an average peak force of 27(15) N during videolaryngoscopy with the GlideScope (Verathon, Inc., Bothell, MA, USA), compared to 39(22) N during direct laryngoscopy with a Macintosh blade [13]. A subsequent clinical trial showed an even greater difference between these blades (8(4) N GlideScope vs. 40(14) N Macintosh) [15]. In a study of patients with features suggestive of difficult direct laryngoscopy, the GlideScope was associated with only a modest 19% reduction in median peak force compared with the Macintosh blade (17 vs. 21 N) [16].

The different blade geometries can be considered as the most relevant factor for the observed differences in force transmission. The curvatures of hyperangulated blades mimic the anatomical shape of the tongue and oropharyngeal airway better. Thus, they reduce the extent of tissue displacement compared to the straighter Macintosh blades. This is despite the more homogeneous spatial distribution of force with the Macintosh blades. With the D-BLADE, the lowest peak and average forces were found in the centre of the sensor array, whereas the maximum forces emerged at the blade’s tip. Other studies confirmed this relationship between curvature and force accumulation at the tip [20,21]. Interestingly, this holds true for both direct and videolaryngoscopy and may explain why we did not observe significant differences between the Macintosh blades. It is of interest that a previous study with manikins described significantly lower peak and impulse force during indirect vs. direct laryngoscopy with a Macintosh-style blade [10]. Here, other factors such as the mechanical properties of the upper airway model and the level of experience of the staff, both difficult to standardize, must be taken into account. As videolaryngoscopy with both the D-BLADE and the Macintosh-style blade is used extensively in our department, the providers in the present study were quite familiar with the use of both types of videolaryngoscopes.

Interestingly, the forces found in the present study were considerably lower than in previous studies with manikins [13,22] and patients [14,15]. It has to be considered, however, that in the present study, the forces were calculated for each sensor specifically. By contrast, previous investigators summed up the forces of all the sensors. Moreover, the forces in manikins often overestimate those performed in humans, as demonstrated by a number of previous trials [13,15]. Finally, but importantly, the method of force measurement should be taken into account in terms of sensor type and position. The shape of a Macintosh blade underlies a common standard and may therefore serve for direct comparison between trials. Carassiti et al. found a peak force of 40(14) N with an effective sensing area of 400 mm^2^ in a size 3 English Macintosh blade [15]. In our test setup, three sensors covered a comparable extension along the blade and thus about 213 mm^2^ of total sensor area. Since force results from the product of pressure and area, the difference in the captured surface might explain the lower forces found in our study. The remaining differences may be, in part, explained by the sensors’ positions. Using a rectangular sensor film, Carassiti et al. included more signals at the very tip of the blade. Thus, it is likely that the overall amount of force found by Carassiti et al. is higher at a comparable extension of the measured area.

Russell et al. used three pressure transducers, similar to ours in type and sensing area [14]. For direct laryngoscopy with a Macintosh blade, they found a mean force of 11(6–16) N and a peak force of 20(14–28) N (both median). Summing up the forces of our first three sensors only would have resulted in 6.3(2.7) N and 13.7(5.7) N, respectively. The reason for these discrepancies may lie in the type and size of the Macintosh blade Russell et al. used. The laryngeal view with the standard Macintosh blade has been demonstrated to be of lower quality compared to the English Macintosh blade used in our study [23]. Thus, one may assume that the provider had to apply more force to achieve sufficient laryngeal view with the standard blades. In regard to hyperangulated blades, a direct comparison is much more difficult. The D-BLADE investigated in the present study is available in either adult or paediatric design. By contrast, the GlideScope comes in different sizes and considerably different geometries. Even though the size of the included model was not reported, from the provided photograph it can be assumed that a small size was used. In contrast to the D-BLADE, the small model of the GlidesScope is flatter in the distal third, and the blade’s tip lies much lower in relation to the base of the handle bar. Thus, the overall angulation of this blade is less distinct, and therefore more force may be required, similar to blades of the Macintosh style.

Our results favour the use of hyperangulated blades in terms of the emerging forces during laryngoscopy. One might conclude that the distribution of force at the blade’s tip may increase the risk of tissue trauma. Our results, however, demonstrate that the absolute forces applied in this location are 40% lower. Such a reduction in force may be clinically relevant in terms of causing less tissue trauma and a reduced risk of oedema and bleeding. A Cochrane review supports these assumptions, having found less frequent laryngeal or airway trauma and lower incidences of postoperative hoarseness after videolaryngoscopy with mainly hyperangulated blades [9]. Interestingly, these advantages seem to prevail, although intubation takes longer with videolaryngoscopy. In this regard, our findings are in accordance with previous research [24].

With respect to clinical impact, the results of our study indicate that hyperangulated blades may be the preferable choice in adults with anatomically normal airways, if forces applied to the airway are to be minimized. Moreover, our findings suggest that techniques of awake intubation under topical anaesthesia may be expanded [25]. Beyond that, the spatially precise assessment of applied forces may help to develop new blades. These could be optimised in terms of material properties and shape, thereby improving ergonomics and avoiding critical incidents [26].

## 5. Limitations of the Study

The experience of the anaesthetist and the patient’s airway status should be considered when interpreting the results. We only included anaesthetists with a certain professional experience, familiar with the devices in question. Forces applied by less experienced providers may exceed those found in our study. Further, we only included patients with expected normal airways. During difficult airway management, laryngoscopy may require more force. However, our results justify the use of hyperangulated blades in patients with normal airways, too, as their use appears to be less traumatic. 

We did not obtain the degree of neuromuscular blockade before laryngoscopy and therefore cannot rule out a certain variability in muscle relaxation. Considering the short duration of otorhinolaryngology procedures, we administered remifentanil for a longer period before propofol was given at a comparatively high dose, followed by cisatracurium. Following this routine, we have had very good experience with regard to the conditions for laryngoscopy in our department. The overall low forces required for laryngoscopy in our study may support these experiences.

A crossover within-subjects study design might have allowed a more accurate comparison of forces between the different blades with fewer patients. However, for reasons of patient risk, an approach involving three intubations in the same patient would not be ethically acceptable. On the other hand, simulating the use of the laryngoscopes without the true intention to intubate may have falsified the results. Therefore, we chose to use a three-arm parallel study design with the respective sample size. 

Our study was conducted with the collaboration of an engineer of the company KARL STORZ (RD). Obtaining a sophisticated picture of force distribution during laryngoscopy necessitated the development of specialised equipment and required technical expertise in its safe use. This was supported via industrial cooperation. The industry-related author prepared the laryngoscopes for the measurements and was involved in the sensor calibrations and hygienic processing of the devices after the measurements. Data collection and analysis were conducted independently of the industry, so we do not believe this study is subject to bias due to industrial cooperation.

## 6. Conclusions

Laryngoscopy with the D-BLADE generated considerably less force on the oropharyngeal tissue compared to blades of the Macintosh style with or without the videolaryngoscopic function. With this knowledge, the study underlines the advantage of hyperangulated blades in terms of risk for tissue trauma in adult patients with normal airways. The spatially precise assessment of applied forces may be useful in the development of new videolaryngoscopic blades, particularly in terms of material properties and shape.

## Figures and Tables

**Figure 1 jcm-13-02623-f001:**
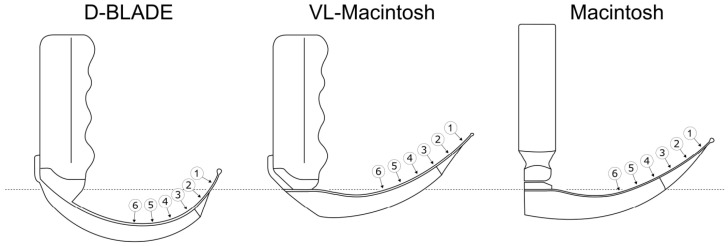
True-to-scale drawings of the laryngoscopy blades used: left to right: C-MAC D-BLADE (D-BLADE; size: adult patient), C-MAC Macintosh blade (VL-Macintosh; size 4) and Macintosh-blade for direct laryngoscopy (Macintosh; size 4), all KARL STORZ SE & Co. KG, Tuttlingen, Germany). For better comparability, the figures are aligned at the base of the blade at the respective handle bar (grey scattered line). Numbers indicate the positions of the tactile force sensors along the blade in the order of the assigned information in the text. Please note the comparatively strong curvature of the D-BLADE and that the sensors cover the blades only partially.

**Figure 2 jcm-13-02623-f002:**
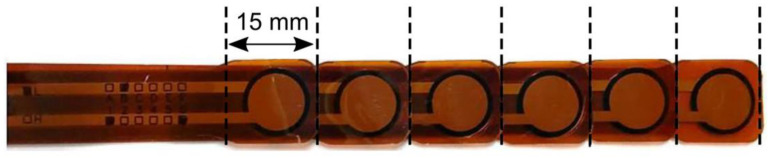
Alignment of the six tactile [1,2,3,4,5,6] force sensors (FlexiForce HT201, Tekscan Inc., Boston, MA, USA) using a medical device adhesive to achieve a combined sensor string. Please note the circular shape of each sensor (diameter 9.5 mm), resulting in an effective sensing area of 71 mm^2^ per sensor.

**Figure 3 jcm-13-02623-f003:**
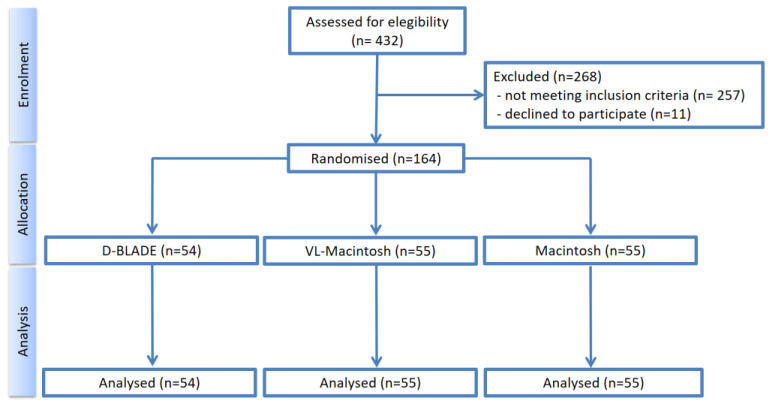
Consort Flow Chart.

**Figure 4 jcm-13-02623-f004:**
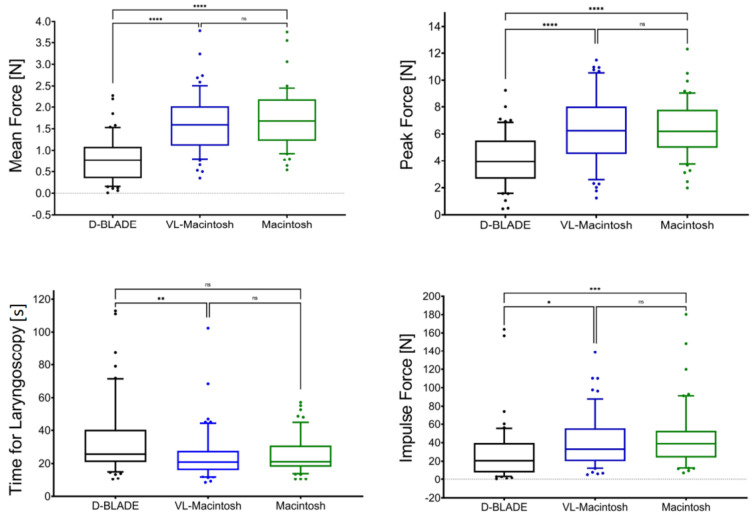
Mean force, peak force, impulse force and time for laryngoscopy during videolaryngoscopy with the C-MAC D-BLADE (D-BLADE; n = 54), videolaryngoscopy with a Macintosh blade (VL-Macintosh; n = 55) and direct laryngoscopy with a Macintosh blade (Macintosh; n = 55). Data are shown as median (IQR 25–75, whiskers from 10 to 90 percentile). ns = not significant, * = *p* < 0.01 ** = *p* < 0.05, *** = *p* < 0.001, **** = *p* < 0.0001.

**Figure 5 jcm-13-02623-f005:**
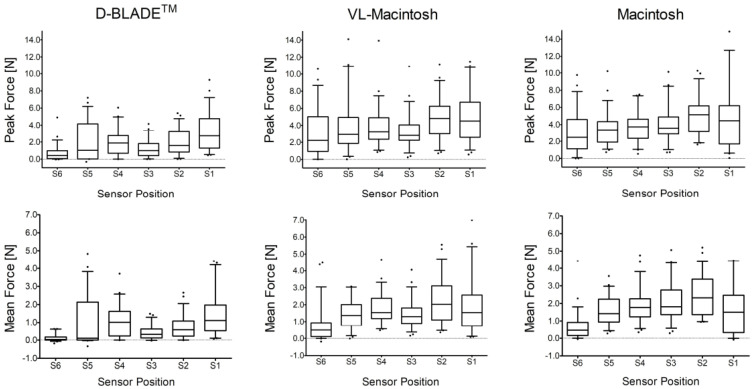
Distribution of peak force (**upper panel**) and mean force (**lower panel**) during videolaryngoscopy with the C-MAC D-BLADE (D-BLADE; n = 54), videolaryngoscopy with a Macintosh blade (VL-Macintosh; n = 55) and direct laryngoscopy with a Macintosh blade (Macintosh; n = 55). The data are shown as box plots (median, IQR 25–75, whiskers from 10 to 90 percentile).

**Figure 6 jcm-13-02623-f006:**
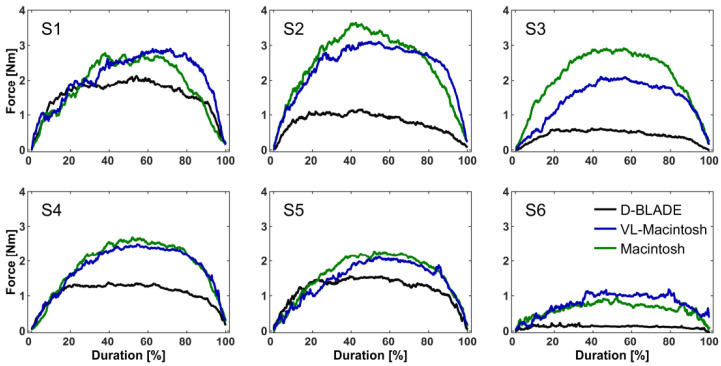
Peak forces acting on isolated force measurement sensors during videolaryngoscopy with the C-MAC D-BLADE (D-BLADE; black lines), videolaryngoscopy with the Macintosh blade (VL-Macintosh; blue lines) and direct laryngoscopy with the Macintosh blade (Macintosh; green lines). S1 refers to the sensor closest to the blade’s tip and S6 closest to the handle (please compare Figure 1). Time intervals are scaled to the total duration of the laryngoscopy in order to ease the comparability of the results. The data are presented as the mean.

**Table 1 jcm-13-02623-t001:** Patients’ characteristics and airway evaluation scores for laryngoscopies with D-Blade and Macintosh laryngoscope blades.

	Videolaryngoscopy		Direct Laryngoscopy
	D-BLADE (n = 54)	VL-Macintosh (n = 55)	Macintosh (n = 55)
Age (y)	55 (45–68)	45 (26–59)	54 (32–68)
Sex Female (n (%)) Male (n (%))	25 (46) 29 (54)	24 (44) 31 (56)	23 (42) 32 (58)
Height (cm)	170 (164–177)	172 (168–178)	173 (168–178)
Weight (kg)	74 (66–85)	74 (60–88)	77 (68–83)
BMI (kg·m^−2^)	25.4 (22.6–29.1)	24.7 (21.8–28.3)	25.2 (23.4–27.4)
Mallampati (I/II/III /IV)	18(33)/21(39)/15(28)/0(0)	23(42)/22(40)/10(18)/0(0)	27(49)/21(38)/7(13)/0(0)
Thyromental dist. (mm)	80 (80–91)	80 (80–91)	80 (70–90)
Interincisor gap (mm)	45 (40–50)	45 (40–50)	47 (40–50)
Arné Risk Index	5 (3–9)	3 (2–8.25)	3 (2–6.25)

Patients’ characteristics and airway evaluation scores for laryngoscopies with D-Blade and Macintosh laryngoscope blades. Data are given as median (25% quartile–75% quartile) or absolute numbers (%). Arné Risk Index: predictive clinical multifactorial risk index for difficult tracheal intubation [18].

## Data Availability

The datasets generated and analysed during the current study are not publicly available due to a large dataset but are available from the corresponding author upon reasonable request.

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
