# Peer review of "Comparison of Force Distribution during Laryngoscopy with the C-MAC D-BLADE and Macintosh-Style Blades: A Randomised Controlled Clinical Trial"

_jcm, 2024, doi:10.3390/jcm13092623_

Round 1

Reviewer 1 Report

Comments and Suggestions for Authors

Thank you for the opportunity to review this manuscript.  I found it very interesting.  The findings reported by the authors are very interesting, how ever, I believe that the manuscript needs to address that the same blades were not used in the same patient.  In my opinion, the study would have been stronger if all 3 blades were used in the same patient, and the force used was reported.  This would have made for a stronger paper as the control, (i.e the patient) would remain the same.  One can then say that there is a true difference in amount of force needed.  Using different blades for different patients, and then concluding less force is needed, I would think one would need a larger sample size.  I hope that the authors can address this limitation.  

Comments on the Quality of English Language

I suggest the manuscript be reviewed again to evaluate sentence structure and overall flow.

Author Response

Thank you for the opportunity to review this manuscript. I found it very interesting. The findings reported by the authors are very interesting, how ever, I believe that the manuscript needs to address that the same blades were not used in the same patient. In my opinion, the study would have been stronger if all 3 blades were used in the same patient, and the force used was reported. This would have made for a stronger paper as the control, (i.e the patient) would remain the same. One can then say that there is a true difference in amount of force needed. Using different blades for different patients, and then concluding less force is needed, I would think one would need a larger sample size. I hope that the authors can address this limitation.

We thank the reviewer for this comment. We agree with the reviewer that evaluating the sample size required to answer our research question is an important step in interpreting the relevance of our results.

Performing three consecutive tracheal intubations in a single patient would not be ethically justifiable, and simulating the use of the laryngoscopes without the true intention to intubate would probably have severely falsified our measurements. Further,, we  planned the sample size according to professional statistical consultation for these conditions.

Therefore we argue that from a statistical point of view that our sample size is large enough to answer our research question appropriately.

However, we briefly discuss reasons for our approach in the limitations section as following:

A crossover within-subjects study design of the study might have allowed a more accurate comparison of forces between the different blades with less patients. However, for reasons of patient risk, an approach involving three intubations in the same patient would not be ethically acceptable. On the other hand, simulating the use of the laryngoscopes without the true intention to intubate may have falsified the results. Therefore, we chose to use a three-arm parallel study design with the respective sample size.

Reviewer 2 Report

Comments and Suggestions for Authors

In the paper "Comparison of force distribution during laryngoscopy with the C-MAC D-BLADE and Macintosh-style blades: a randomized controlled clinical trial", the authors compared the force distribution with three different laryngoscopy blades.

My comments:

- Please specify the units of measurement in the summary

- Table 1. - please specify a statistical method for between-group comparisons. 

- Did the groups differ in certain demographic characteristics?

 Figure 4. - Unit for Time for laryngoscopy -  is it second or N?

I have no other comments 

Author Response

In the paper "Comparison of force distribution during laryngoscopy with the C-MAC D-BLADE and Macintosh-style blades: a randomized controlled clinical trial", the authors compared the force distribution with three different laryngoscopy blades.

My comments:

- Please specify the units of measurement in the summary

We checked our manuscript for missing units, but didn't find any. We concluded that the reviewer may be referring to the abstract, where we do not provide numerical values for forces, but only comparative information on our results, which are presented independently of the unit. Therefore, we did not change the manuscript in this respect.

- Table 1. - please specify a statistical method for between-group comparisons. - Did the groups differ in certain demographic characteristics?

We agree with the reviewer that randomisation is an important issue. However, we did not perform statistical tests on patient characteristics. First, hypothesis tests are generally not suitable for the evaluation of similarity. Second, regarding the randomization, the null hypothesis that the two groups come from the same population are generally true and a significant difference would have occurred per definition by chance. With regard to the stratified randomisation approach, we consider it justified to assume adequate randomisation.

 Figure 4. - Unit for Time for laryngoscopy -  is it second or N?

We thank the reviewer for this comment and changed the unit into seconds [s].

Reviewer 3 Report

Comments and Suggestions for Authors

Overall, this manuscript is good. There are only a few comment for this. Please see the attachment.

Author Response

Overall, this manuscript is good. There are only a few comment for this.

In Table 1, It is mentioned Arne Risk Index (Line 187), but the Arne Risk Index was not mentioned in the method section. Please add an explanation regarding Arne Risk Index and how ot interpret it in the Materials and Methods section.

We thank the reviewer for this comment and added the Arne Risk Index Score in the Methods section including an explanation as following: 

Patients´ characteristics and airway evaluation including Arne risk index score [19], a multivariable risk index score for the preoperative assessment of difficult intubation with direct laryngoscopy, were recorded. An Arne risk index score of 11 or greater identifies difficult tracheal intubation with a sensitivity of 94 per cent and specificity of 96 per cent in general surgery patients.

There were incomplete writing in Table 1. In column 2-4, It is mention

  1. Age : 55 (45-64)   45 (26-59)     54 (32-68)

What is the meaning of number in the column (yellow block)? Is it (min-max) or range? Please clarify it.

  1. This also applies to Height, Weight, BMI, Thyromental dist., Inter-incisor gap, and Arne Risk Index.
  2. As a suggestion in column 1, it can be write as

“Age, years (range) or Age, years (min-max)”

We thank the reviewer for this comment. We added the information in the table legend as following:
Data are given as median (25% quartile - 75% quartile) or absolute numbers (%)

Reviewer 4 Report

Comments and Suggestions for Authors

This is a well-performed clinical trial providing important insights on how different types of laryngoscope blades affect the pressure on pharyngeal tissues, thereby contributing to the discussion on the differences between direct and videolaryngoscopic intubation techniques. The presentation and discussion of the results is exceptional. Congratulations for conducting this interesting trial!

While I don't see major issues, some minor comments that should be addressed prior to publication in the following:

-  Your initial statements on the significance of videolaryngoscopy and the comparison with direct methods could be strengthened by citing the most recently published large trials, comparing direct laryngoscopy with videolaryngoscopy for intubation. These clearly showed the advantage of videolaryngoscopy over direct laryngoscopy in various clinical settings.

Examples, but not exclusive:

Kriege, M., Noppens, R.R., Turkstra, T., Payne, S., Kunitz, O., Tzanova, I., Schmidtmann, I. and (2023), A multicentre randomised controlled trial of the McGrath™ Mac videolaryngoscope versus conventional laryngoscopy. Anaesthesia, 78: 722-729. https://doi.org/10.1111/anae.15985

Ruetzler KBustamante SSchmidt MT, et al. Video Laryngoscopy vs Direct Laryngoscopy for Endotracheal Intubation in the Operating RoomA Cluster Randomized Clinical TrialJAMA. 2024;331(15):1279–1286. doi:10.1001/jama.2024.0762

Prekker ME, Driver BE, Trent SA, Resnick-Ault D, Seitz KP, Russell DW, Gaillard JP, Latimer AJ, Ghamande SA, Gibbs KW, Vonderhaar DJ, Whitson MR, Barnes CR, Walco JP, Douglas IS, Krishnamoorthy V, Dagan A, Bastman JJ, Lloyd BD, Gandotra S, Goranson JK, Mitchell SH, White HD, Palakshappa JA, Espinera A, Page DB, Joffe A, Hansen SJ, Hughes CG, George T, Herbert JT, Shapiro NI, Schauer SG, Long BJ, Imhoff B, Wang L, Rhoads JP, Womack KN, Janz DR, Self WH, Rice TW, Ginde AA, Casey JD, Semler MW; DEVICE Investigators and the Pragmatic Critical Care Research Group. Video versus Direct Laryngoscopy for Tracheal Intubation of Critically Ill Adults. N Engl J Med. 2023 Aug 3;389(5):418-429. doi: 10.1056/NEJMoa2301601. Epub 2023 Jun 16. PMID: 37326325.

- Please add commonly used subchapters to the methods section, such as study design, study population, measurements, outcomes, statistics, etc... (see CONSORT checklist as a reference).

- Please make sure this manuscript in total adheres to the CONSORT guideline and state so within the manuscript.

- Please add what statistical measures you used to the legend of table 1.

- Is there any previous direct evidence how the applied force to pharyngeal tissue correlates with injuries or patient discomfort after intubation? If so, please add this information to the discussion or introduction section.

Author Response

This is a well-performed clinical trial providing important insights on how different types of laryngoscope blades affect the pressure on pharyngeal tissues, thereby contributing to the discussion on the differences between direct and videolaryngoscopic intubation techniques. The presentation and discussion of the results is exceptional. Congratulations for conducting this interesting trial!

Thank you for this assessment.

While I don't see major issues, some minor comments that should be addressed prior to publication in the following:

- Your initial statements on the significance of videolaryngoscopy and the comparison with direct methods could be strengthened by citing the most recently published large trials, comparing direct laryngoscopy with videolaryngoscopy for intubation. These clearly showed the advantage of videolaryngoscopy over direct laryngoscopy in various clinical settings.

Examples, but not exclusive:

Kriege, M., Noppens, R.R., Turkstra, T., Payne, S., Kunitz, O., Tzanova, I., Schmidtmann, I. and (2023), A multicentre randomised controlled trial of the McGrath™ Mac videolaryngoscope versus conventional laryngoscopy. Anaesthesia, 78: 722-729. https://doi.org/10.1111/anae.15985

Ruetzler K, Bustamante S, Schmidt MT, et al. Video Laryngoscopy vs Direct Laryngoscopy for Endotracheal Intubation in the Operating Room: A Cluster Randomized Clinical Trial. JAMA. 2024;331(15):1279–1286. doi:10.1001/jama.2024.0762

Prekker ME, Driver BE, Trent SA, Resnick-Ault D, Seitz KP, Russell DW, Gaillard JP, Latimer AJ, Ghamande SA, Gibbs KW, Vonderhaar DJ, Whitson MR, Barnes CR, Walco JP, Douglas IS, Krishnamoorthy V, Dagan A, Bastman JJ, Lloyd BD, Gandotra S, Goranson JK, Mitchell SH, White HD, Palakshappa JA, Espinera A, Page DB, Joffe A, Hansen SJ, Hughes CG, George T, Herbert JT, Shapiro NI, Schauer SG, Long BJ, Imhoff B, Wang L, Rhoads JP, Womack KN, Janz DR, Self WH, Rice TW, Ginde AA, Casey JD, Semler MW; DEVICE Investigators and the Pragmatic Critical Care Research Group. Video versus Direct Laryngoscopy for Tracheal Intubation of Critically Ill Adults. N Engl J Med. 2023 Aug 3;389(5):418-429. doi: 10.1056/NEJMoa2301601. Epub 2023 Jun 16. PMID: 37326325.

We thank the reviewer for this comment and added the most recently published large trials documenting the advantage of videolaryngoscopy over direct laryngoscopy for tracheal intubation in the introduction section.

This forecast is supported by accumulating evidence, attributing an improved view of the larynx [2], lower incidence of postoperative sore throat [3], improved first pass success [Kriege M, EMMA trial 2023][Prekker ME, NEJM 2023][Kriege M, LARA trial 2024], decreased number of attempts needed [Ruetzler K, JAMA 2024] and a great teaching potential to videolaryngoscopy [4].

- Please add commonly used subchapters to the methods section, such as study design, study population, measurements, outcomes, statistics, etc... (see CONSORT checklist as a reference).

We thank the reviewer for this comment and added the subchapters to the methods section.

- Please make sure this manuscript in total adheres to the CONSORT guideline and state so within the manuscript.

We thank the reviewer for this comment and included the adherence to the CONSORT guideline in the methods section.

- Please add what statistical measures you used to the legend of table 1.

We thank the reviewer for this comment and added the statistical measures to the legend of table 1.

Data are given as median (25% quartile - 75% quartile) or absolute numbers (%)

- Is there any previous direct evidence how the applied force to pharyngeal tissue correlates with injuries or patient discomfort after intubation? If so, please add this information to the discussion or introduction section.

The reviewer raises an important point. Although there are data (see below) on soft tissue trauma (most commonly to the tongue) after laryngoscopy and tracheal intubation, a correlation between tissue trauma and the force applied has not been reported, to our knowledge yet.

Mourão J, Moreira J, Barbosa J, Carvalho J, Tavares J. Soft tissue injuries after direct laryngoscopy. J Clin Anesth. 2015 Dec;27(8):668-71.

Komasawa N, Komatsu M, Yamasaki H, Minami T. Lip, tooth, and pharyngeal injuries during tracheal intubation at a teaching hospital. Br J Anaesth. 2017 Jul 1;119(1):171